# In-context Prompt Learning for Test-time Vision Recognition with Frozen Vision-language Model

## Abstract

Existing pre-trained vision-language models, *e.g.*, CLIP (Radford et al., 2021), have demonstrated impressive zero-shot generalization capabilities in various downstream tasks. When facing test inputs with different distributions, however, the performance of these models will degrade significantly. To this end, we explore the concept of test-time prompt tuning (TTPT), which enables the adaptation of the CLIP model to novel downstream tasks through only one step of optimization on an unsupervised objective that involves the test sample. One noteworthy aspect is that TTPT operates without specific task supervision, such as fine-tuning. This makes it a particularly intriguing approach, as it allows us to investigate how a pre-trained visual-language model like CLIP can adapt to downstream tasks without requiring task-specific supervision or model modifications.

Drawing inspiration from recent advancements in in-context learning within the field of natural language processing (NLP), we introduce the concept of *visual in-context prompting*. This involves associating a new test sample with very few or even just one labeled example as its in-context prompt. As a result, we can reliably estimate a label for the test sample, thereby facilitating the adaptation process. Our approach employs a token net to represent language descriptions as visual prompts that the vision encoder of a CLIP model can comprehend. Paired with in-context examples, we further propose a semi-supervised loss to optimize test sample-aware visual prompts. This optimization allows a pre-trained, frozen CLIP model to be adapted to a test sample from any task using its learned adaptive prompt. To further enhance the integration of visual and text prompts, we design a cyclic learning strategy. Our method has demonstrated superior performance and achieved state-of-the-art results across various downstream datasets.

## 1 Introduction

Recent advances in vision-language pre-training, such as CLIP (Radford et al., 2021), have shown a promising direction for developing foundation models for downstream tasks (Bommasani et al., 2021). These foundation models are trained on extensive web-scale data, such as 400 million text-image pairs in the case of CLIP, to align language and vision modalities. When transferring to downstream tasks, the pre-trained model needs to be fine-tuned on a few labeled images for the task domain. Fully fine-tuning a large pre-trained model (Bommasani et al., 2021; Liu et al., 2021a; Cai et al., 2022) for each downstream task faces two major challenges in real-world applications. One practical concern is the storage and distribution issue, as maintaining a separate model copy for each task is costly and inflexible, particularly with an expanding number of downstream tasks. Another one is that fully fine-tuning has large probability to destroy initial knowledge provided by the large-scale pre-trained model, increasing the risk of model overfitting.

Unlike the fully fine-tuning to expensively update all model parameters for each downstream task, prompt tuning methods (Zhou et al., 2022b;a) prepend the inputs with learnable parameters that steers the foundation model towards generating the desired outputs. Recent works (Zhou et al., 2022b;a; Bahng et al., 2022; Jia et al., 2022) use training data to tune prompts since the embeddings of prompt are extracted from the model input and are differentiable with respect to the loss function. However, the learned prompts are usually limited to the training data, and their performance

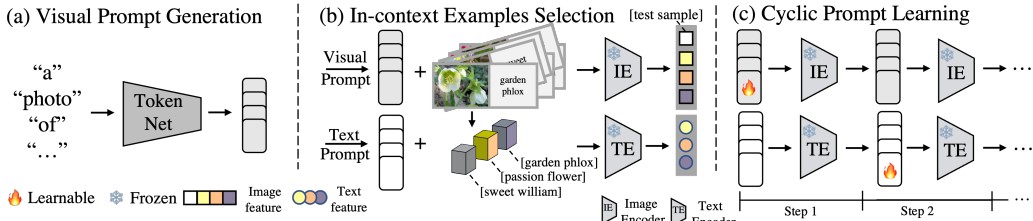

Figure 1: Illustration of the proposed in-context prompt learning (InCP) framework. Our approach focuses on identifying unknown objects within a contextual setting using learned prompts, all without the need for extensive fine-tuning of a large pre-trained model. See texts for details.

tend to degrade significantly when the test samples are drawn from a different distribution. Test-time prompt tuning (TTPT) offers a solution to the distribution problem through one-step optimization on an unsupervised objective that involves the unlabelled test samples. Such flexibility of TTPT allows to address the distribution shift by prompt tuning in a zero-shot manner. For example, TPT (Shu et al., 2022) exploit the potential of pre-trained vision-language foundation models as better zero-shot learners by maximizing the consistency of predictions across augmented versions of each test sample. However, the learned prompt via existing TTPT methods might be noisy and irrelevant to task-specific distribution. This raises the question: How can we effectively tailor a pre-trained visual-language model (e.g., CLIP) to a new task by judiciously incorporating domain-specific information during testing?

Recent works in natural language processing (NLP) have shown that in-context learning (ICL) [1] (Brown et al., 2020) are effective in adapting foundation models for downstream tasks including knowledge retrieval (Tay et al., 2022; Wang et al., 2020) and multi-choice tasks (Min et al., 2022). Leveraging In-Context Learning (ICL), large language models can tackle novel tasks solely through inference, conditioning on a limited set of input-label pairs to make predictions for new inputs. To enhance the adaptation of pre-trained vision-language models to new tasks, we introduce an innovative approach based on ICL. This approach prompts the model to acquire domain-specific context information for a test sample using just a few in-context examples, which are derived from domain-specific input-label pairs.

In this paper, we introduce visual In-Context Prompt learning (InCP), a method that empowers a pre-trained vision-language model, such as the CLIP model, to leverage in-context examples. Our approach maintains the frozen weights of the vision-language model while enabling gradient backpropagation through it. This allows us to dynamically fine-tune the visual prompt using in-context examples and an unlabeled test sample. To achieve this, we undertake the following steps, as illustrated in Figure 1: (*i*) We employ a token network to convert textual descriptions into a vision prompt that the vision encoder can comprehend (Figure 1 (a)). (*ii*) We construct a test sample coupled with in-context candidates to harness the capabilities of the pre-trained large model (Figure 1 (b)). (*iii*) We design an effective learning strategy to seamlessly integrate the visual prompt with the text prompt (Figure 1 (c)). Through these steps, InCP enables the CLIP model to adapt to new tasks by leveraging in-context information, all while keeping the model's core parameters fixed. By harnessing the power of a pre-trained vision-language model, InCP inherits the robust zero-shot learning capabilities of the CLIP model for downstream tasks on which it was not initially trained. During testing, InCP seamlessly adapts to new tasks with the assistance of just a few in-context examples and a test sample. Finally, we conduct our experiments on diverse image recognition datasets and evaluate the transfer performance with pre-trained models. Our InCP outperforms previous methods by a large margin, especially on fine-grained datasets. We also provide comprehensive ablation study and qualitative analysis to understand the effectiveness of InCP.

Our main contributions can be summarized as follows: 1) We introduce InCP, a straightforward yet highly effective method for enhancing the CLIP model with in-context examples. This marks the first application of visual in-context prompt learning across a range of downstream tasks. 2)

---

[1]Without the need to fine-tune any model parameters for downstream tasks, in-context learning involves adding domain-specific input-output pairs, referred to as in-context examples, to a test example. This prompts the model to learn relevant patterns for the test sample, enabling it to perform multiple tasks seamlessly.

We delve into the effective representation of language descriptions as visual prompt initialization. This involves the utilization of in-context examples and query samples to harness the potential of the vision encoder within the CLIP model. Additionally, we design a learning strategy for visual demonstrations that seamlessly integrate visual prompts with text prompts. 3) Our InCP approach is rigorously evaluated through extensive experiments, demonstrating superior performance and achieving state-of-the-art (SOTA) results across a diverse set of downstream datasets.

## 2 RELATED WORK

**Prompt Learning in Vision-Language models.** Foundational vision-language (V-L) models (Radford et al., 2021; Jia et al., 2021; Zhai et al., 2022) exploit both visual and textual modalities to encode multi-modal representations. These models are pre-trained a large amount of image-text pairs available online in a self-supervised manner. During pre-training stage, a contrastive loss function is used to pull together the features of paired images and texts while pushing away the unpaired image-text features. V-L models like CLIP (Radford et al., 2021), FILIP (Yao et al., 2021) and Florence (Yuan et al., 2021) have demonstrated impressive zero- and few-shot generalization capabilities in a wide range of downstream tasks. Inspired by prompt learning in natural language process (NLP), recent works have proposed to adapt V-L models to downstream task by learning the prompt tokens. CoOp (Zhou et al., 2022b) prepends a category name with a prompt "a photo of a" (e.g., "a photo of a cat"), and optimizes continuous set of prompt vectors at its language branch. CoCoOp (Zhou et al., 2022a) further makes text prompt conditioned on each input instance (image) and thus enable dynamic text prompt to adapt to each instance. As a visual prompt technique, VP (Bahng et al., 2022; Jia et al., 2022) introduces additional random noises as task-specific learnable parameters into input image space. Their prompts are exclusively concentrated within the visual branch. Afterwards, MaPLe (Khattak et al., 2023) extend CoOp (Zhou et al., 2022b) and VP (Bahng et al., 2022) to effectively leveraging two kinds of prompts to ensure synergy between vision-language modalities during training stage. However, the learned prompts are limited to the training data distribution and their performances significantly degrade when the model is tested on inputs drawn from a different distribution. To fully delve into the utilization of visual prompts for adapting V-L models, we explore a novel approach based on *in-context* learning (Brown et al., 2020) to strongly encourage V-L model adapt to new tasks quickly by using only few in-context examples.

**Test-time Adaptation Methods.** Test-time adaptation (TTA) (Goyal et al., 2022) focus on challenging scenarios with the presence of only a source model and unlabeled target data are available. This work primarily attempts to address the distribution shift by designing effective test-time objective about test sample. Two similar paradigms are test-time training (TTT) (Sun et al., 2020; Liu et al., 2021b) and domain adaptation (DA) (Saito et al., 2019; Tzeng et al., 2015). TTT adds a self-supervised proxy task, such as recognizing rotations of an image, to training objective during training stage and computes an optimization objective at test time, while DA requires the utilization of both source and target data for training with a cross-domain loss function. Without source training data, TTA (Shin et al., 2022) does not require joint training across losses (TTT) or domain adaptation (DA). The above settings usually needs to fine-tune specific model parameters, which may discard valuable knowledge provided by the pre-trained model and increase the risk of model overfitting. TPT (Shu et al., 2022) is the first test-time prompt tuning work to enhance the zero-shot generalization of CLIP model by learning adaptive prompts for each test sample at test time. However, TPT focus only on unlabeled test sample, which leads to the prompted (retrieved) knowledge features from CLIP model might deviate from their original meanings within the target distribution.

**In-context Learning.** In-context learning (ICL) defined by GPT3 (Brown et al., 2020) is a new paradigm, where autoregressive language model can perform on-the-fly computational reasoning on unseen tasks (Min et al., 2021) given prompts and examples serving as context. Flamingo (Alayrac et al., 2022) extends the concept of ICL to images and videos by employing language as instruction. Wang et al. (2023a) conditions the input on a pair of input-output images from the same task to indicate which task to perform. Zhang et al. (2023) further explore the impact of incontext examples for large vision models. These works (Wang et al., 2023a; Zhang et al., 2023) are developed for visual dense prediction but is not applicable to vision and language (V-L) model. Our work follows them but studies visual in-context learning from a different dimension: how to learn prompt for test-time V-L model adaptation with test sample and in-context examples.

# 3 METHOD

## 3.1 PRELIMINARIES: TEST-TIME PROMPT LEARNING

**CLIP Models.** In this paper, we focus on generalizing the vision-language pre-trained model (CLIP) (Radford et al., 2021) to downstream tasks in zero-shot manner while keeping the model parameters frozen. Without supervision from human labels, CLIP can directly learn associated visual-text representations from a large amount of image-text pairs. Specifically, CLIP trains a vision encoder $g_I(\cdot)$ and a text encoder $g_T(\cdot)$ in contrastive learning to align the embeddings and then match image-text pairs. During testing, zero-shot recognition is accomplished by comparing image features with the class names synthesized by the text encoder.

**Test Time Prompt learning.** With pre-trained CLIP model, test time prompt learning approaches (Shu et al., 2022) append learnable prompt tokens at the text encoder during test time, e.g., "a photo of a" is added to every class names to form "a photo of a {class}". This provides the model very helpful context information about the downstream task. Formally, class label is wrapped within a language prompt, which can be formulated as $\{t_{SOS}, P_t, c, t_{EOS}\}$. Here, $c$ represents the class label, and $P_t = \{P_l\}_{l=1}^4$ denotes the learnable prompt initialized with word embeddings corresponding to "a photo of a". $t_{SOS}$ and $t_{EOS}$ are the learnable start and end token embeddings.

**Problem Definition.** Given a pre-trained model $f_{\theta_0}$ with parameters $\theta_0$ and prompt $P$, the proposed In-context Test-Time Adaptation (ITTA) aims to adapt $f_{\theta_0}$ with only few input-label pairs (known as in-context examples) $(x_i, y_i)$ and test samples $x_t$ to downstream task. During testing, ITTA fine-tunes a learnable prompt using an optimization objective. This process enhances the model's generalizability for downstream tasks where test samples lack labels, but in-context examples are accessible. The candidate in-context example set is formed by selecting one image-label pair from each category, followed by random sampling of several pairs from this set to create in-context examples. For a more comprehensive discussion on the selection of examples, we refer to the Section A.2 in the Appendix. Without requiring source data $(x_s, y_s)$, ITTA does not require joint training across domains (domain adaptation) or losses (test time training). In contrast to test-time adaptation, ITTA can provide additional in-context examples as domain-specific context information for target distribution. The difference of ITTA and other adaptations are shown in Table 1.

Table 1: Characteristics of in-context test-time adaptation (ITTA) and other adaptations.

| Setting | Adaptation | | Available Data | | | Loss | |
|---|---|---|---|---|---|---|---|
| | Train | Test | Source | Target | In-context | Train loss | Test loss |
| Fine-tuning | ✓ | × | - | $x_t, y_t$ | - | $L(x_t, y_t)$ | - |
| Domain Adaptation | ✓ | × | $x_s, y_s$ | $x_t$ | - | $L(x_s, y_s) + L(x_s, x_t)$ | - |
| Domain Generalization | ✓ | × | $x_s, y_s$ | | - | $L(x_s, y_s)$ | - |
| Test-Time Training | ✓ | ✓ | $x_s, y_s$ | $x_t$ | - | $L(x_s, y_s) + L(x_s)$ | $L(x_t)$ |
| Fully Test-Time Adaptation | × | ✓ | - | $x_t$ | - | - | $L(x_t)$ |
| ITTA | × | ✓ | - | $x_t$ | $x_i, y_i$ | - | $L(x_t) + L(x_i, y_i)$ |

## 3.2 VISUAL PROMPT INPUT WITH IMAGES

The visual prompt approach enables the learning of image perturbations (such as noisy boxes or rectangles) for a frozen model. This allows the model, when prompted with these perturbations, to perform a new task. As depicted in Figure 3 (a) and (b), recent studies (Bahng et al., 2022; Jia et al., 2022) have introduced task-specific learnable parameters as visual prompts. These prompts are prepended into the input image space and are learned during the fine-tuning stage. However, these parameters are typically initialized with random noise. We contend that a general text description (e.g., a photo caption) provides richer visual context information compared to random noise. This is because linguistic words can be perceived as interpretable lexical tokens that can be easily translated into visual content comprehensible to the vision encoder. This is especially advantageous during the test-time adaptation.

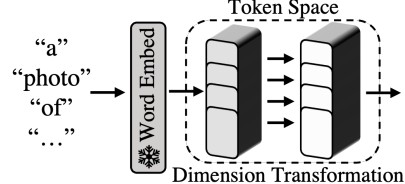

Figure 2: The illustration of token net.

A straightforward approach to implementing a visual prompt is to combine patch image embedding with learnable parameter vectors initialized with word embeddings. However, this design not only lacks a seamless connection between the vision and language modalities but also faces the challenge of differences in dimensionality between these two types of embeddings. In this regard, we propose

an efficient language-to-vision translation design that has proven to be highly effective in practice. Specifically, in addition to the word context vectors, we introduce the concept of a lightweight neural network, known as Token-Net. Token-Net is responsible for generating a conditional token vector for each input image, which is then integrated with the context visual vectors. For a visual representation of this architecture, please refer to Figure 2.

Let $\mathcal{F}_\theta$ denote token net parameterized by $\theta$, each visual prompt $\boldsymbol{P_v}$ is obtained by projecting text words in a prompt $\boldsymbol{P_t}$ into a set of learnable vectors via language-to-vision token net, such that $\boldsymbol{P_v} = f_\theta(\boldsymbol{P_t})$. Given as input image $\mathbf{X} \in \mathbb{R}^{C \times H \times W}$, we divide it into $M$ pathes and produce patch tokens through a projection at vision branch of CLIP model. The visual prompt for the $i$-th input image is thus conditioned on the language description. The token net is built with a linear layer, which maps inputs of dimensions $d_l$ to $d_v$. We leave the exploration of more advanced designs to future endeavors. During training, we update the context vectors $\boldsymbol{P_v}$ together with Token-Net's parameters $\theta$. This acts as a bridge between visual and language modalities, thus reducing modality discrepancy and encouraging knowledge transfer from language to vision. Unlike CoCoOP (Zhou et al., 2022a), which solely conditions a text prompt on each input image without the learning of visual prompts, the explicit conditioning of our visual prompt on natural language ensure the initial visual prompts within a meaningful embedding space, thereby facilitating a smoother convergence process.

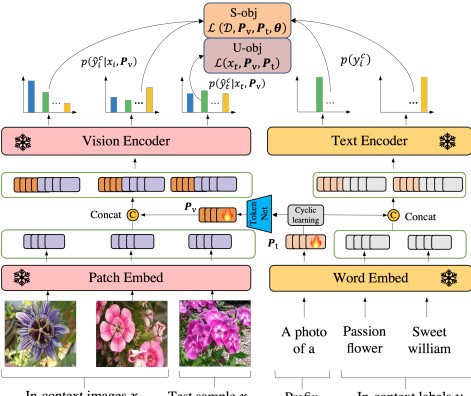

Figure 3: The distinction between our language-aware visual prompt approach and the patched, padded, and token-based visual prompt methods.

## 3.3 IN-CONTEXT LEARNING FOR VISUAL RECOGNITION

In-context learning is an emerging paradigm that involves supplying labeled examples as input to a pre-trained language model. Alongside these examples, a query example is provided for which the model must predict the label based on its understanding of the task gleaned from the provided examples. Unlike conventional few-shot learning methods, in-context learning does not necessitate the updating of all weight parameters of the pre-trained model or the learning of task-specific prompts with training data for each task. Our method, InCP, extends this approach into the realm of visual recognition. It does so by conditioning visual prompt learning on task-specific image-label pairs while keeping the model parameters frozen. As a result, our proposed method eliminates the need for customizing models for individual tasks and generates sample-aware prompts for the foundational model. The illustration of the proposed InCP is shown in Figure 4.

Formally, given an unlabeled test sample $x_t$ and some in-context examples $\mathcal{D} = \{x_i, y_i\}_{i=1}^N$ containing $N$ image-label pairs (i.e., an image and its class name), in-context learning can be formulated as

$$y_t = \mathcal{F}(\mathcal{D}, x_t; \Phi). \tag{1}$$

In-context examples $\mathcal{D}$ offers contextual guidance (prompt) to the model $\mathcal{F}(\cdot; \Phi)$, enabling it to generate the optimal $y_t$ for $x_t$ without updating model

Figure 4: Illustration on the proposed visual in-context prompt learning for test-time visual recognition. Each in-context example $(x_i, y_i)$, test sample $x_t$, and its prefix text are fed into a token encoder to obtain visual, prefix, and text tokens. The text tokens are translated into a visual prompt using a token network. We optimize the visual tokens using a semi-supervised objective: a supervised cross-entropy term $L(x_i, y_i, \boldsymbol{P_v})$ involving in-context examples and an unsupervised entropy minimization $L(x_t, \boldsymbol{P_v})$ with the test sample.

parameters. As a kind of unsupervised prompt learning (Zhang et al., 2023), in-context learning relies heavily on the pre-trained model and its objective may well not align with that used in pre-training. Our InCP append visual prompts given as $\boldsymbol{P_v}$ with the visual tokens $\{e_{cls}, e_1, e_2, \ldots, e_M\}$. Hence, the visual encoder processes the following input tokens $\widetilde{X}_p = \{\boldsymbol{P_v}, e_{cls}, e_1, e_2, \ldots, e_M\}$ to produce the prompted visual feature represented as $f_{\widetilde{p}} = \mathcal{F}(X_p; \Phi)$. Then, the instance-specific visual prompt is learned on the following objective to provide the model with the most helpful context information about on-the-fly test sample $x_t$.

**Test-Time Objective.** Given that sample's label is not available for test-time adaptation, we propose a semi-supervised loss for visual in-context prompt learning. At each time step of adaptation, we comprise a batch of test inputs with on-the-fly test sample and in-context examples. For unlabeled test sample $x_t$, our test-time unsupervised objective (U-obj) is to minimize the Shannon entropy (Shannon, 1948), $L(x_t, \boldsymbol{P_v}) = -\sum_c p(\hat{y}^c) \log p(\hat{y}^c | x_t, \boldsymbol{P_v})$ for the model predictions $\hat{y} = f_\theta(x_t)$ of class $c$. Given image-label pairs $\mathcal{D} = \{x_i, y_i\}_{i=1}^N$, our test-time supervised objective optimize the same loss on in-context examples with a supervised loss (S-obj) $L(x_i, y_i, \boldsymbol{P_v}) = -\sum_c p(y_i) \log p(\hat{y}_i^c | x_i, \boldsymbol{P_v})$. We take *a gradient step* towards optimizing the semi-supervised loss $L = L(x_t, \boldsymbol{P_v}) + \sum_{i=1}^N L(x_i, y_i, \boldsymbol{P_v})$ over learnable parameters $\boldsymbol{P_v}$ that are shared across the test batch. The overall learning objective is formulated as follows:

$$\boldsymbol{P_v^*} = arg \min_{\boldsymbol{P_v}} \left\{ L(x_t, \boldsymbol{P_v}) + \sum_{(x_i, y_i) \in \mathcal{D}} \lambda L(x_i, y_i, \boldsymbol{P_v}, \theta) \right\}. \tag{2}$$

Here $\lambda$ is loss weight parameter balancing the contributions of different loss components. Each of these samples only necessitates a single pass through the visual-language model, coupled with a shared prompt. Following a single-step optimization process involving the semi-supervised objective, which incorporates on-the-fly test samples, we assess the online performance of the proposed InCP method on these test samples, utilizing the learned prompt. In the end, we utilize the average model performance on the testing data as the evaluation metric.

### 3.4 CYCLIC LEARNING OF VISUAL AND LANGUAGE PROMPTS

Visual and language (V-L) prompts are two parameter-effective prompt-tuning methods. Most methods (Khattak et al., 2023; khattak et al., 2023) attempt to simultaneously adapt vision and language branch of CLIP model with learnable prompts of both modalities, which ensures mutual synergy between V-L prompts. This mutual learning strategy performs effectively when training data provides alignment information across visual and language modalities. However, this performance is limited for the target distribution during test-time adaptation, where on-the-fly test sample is unlabeled (i.e., only visual information is available). To this end, we propose a cyclic learning of visual and language prompt to tune vision and language branch of CLIP sequentially, which leverages context information obtained from one-shot examples. Specifically, we start by optimizing one prompt and subsequently utilize the knowledge gained from this step to guide the optimization of the remaining prompt. This process provides the model with richer contextual information, allowing it to thoroughly capture relevant patterns of on-the-fly test sample to align visual and language modalities. The optimization objective for the proposed cyclic learning strategy can be formulated by

$$\mathcal{L}(x_t, \mathcal{D}, \boldsymbol{P_v}, \boldsymbol{P_t}, \theta) = \begin{cases} arg \min_{\boldsymbol{P_v}, \theta} \left\{ L(x_t, \boldsymbol{P_v}) + \sum_{(x_i, y_i) \in \mathcal{D}} \lambda L(x_i, y_i, \boldsymbol{P_v}, \boldsymbol{P_t}, \theta) \right\}, s = 1, \\ arg \min_{\boldsymbol{P_t}} \left\{ L(x_t, \boldsymbol{P_t}) + \sum_{(x_i, y_i) \in \mathcal{D}} \lambda L(x_i, y_i, \boldsymbol{P_v}, \boldsymbol{P_t}, \theta) \right\}, s = 2, \end{cases}$$
$$\tag{3}$$

where $x_t$ is unlabeled test sample, $\mathcal{D}$ contains in-context examples, $\theta$ is the parameter of token net, and $s = 1, 2$ are the step numbers of model optimization for visual and text prompt, respectively. It is noted that the paragraphs above section 3.4 are focused on visual prompt learning, which is a single-step optimization process. In this section, we introduce text prompts to integrate visual prompts. This process is a two-step optimization process, where each type of prompt is optimized in sequence, i.e., first visual then textual prompts. It forms the "cyclic prompt learning" concept.

## 4 EXPERIMENTS

**Datasets.** Here we evaluate the transfer performance of our method on 9 fine-grained classification datasets and 4 out-of-distribution (OOD) data of ImageNet. Fine-grained includes specific species

Table 2: Accuracy comparison with previous methods on fine-grained classification datasets. CoOp and CoCoOp are fine-tuned on the ImageNet dataset using 16-shot training data per category. Baseline CLIP, prompt ensemble, and TPT do not require ImageNet dataset as training data. Our method builds above TPT and further learns sample-specific prompts for test sample by using only few in-context examples. The top-1 classification accuracy is reported on each dataset.

| Method | Type | Flower | DTD | Pets | Cars | UCF101 | Caltech | Food | Aircraft | EuroSAT | SUN | Average |
|---|---|---|---|---|---|---|---|---|---|---|---|---|
| CLIP-ViT-B/16 | Zero-shot | 67.44 | 44.27 | 88.25 | 65.48 | 65.13 | 93.35 | 83.65 | 23.67 | 42.01 | 62.59 | 63.58 |
| Ensemble (Shu et al., 2022) | Zero-shot | 66.99 | 45.04 | 86.92 | 66.11 | 65.16 | 93.55 | 82.86 | 23.22 | 50.42 | 65.63 | 64.59 |
| TPT (Shu et al., 2022) | Zero-shot | 68.98 | **47.75** | 87.79 | 66.87 | 68.04 | 94.16 | **84.67** | 24.78 | 42.44 | 65.50 | 65.10 |
| CoOp (Zhou et al., 2022b) | Few-shot | 68.71 | 41.92 | 89.14 | 64.51 | 66.55 | 93.70 | 85.30 | 18.47 | 46.39 | 64.15 | 63.88 |
| CoCoOp (Zhou et al., 2022a) | Few-shot | 70.85 | 45.45 | 90.46 | 64.90 | 68.44 | 93.79 | 83.97 | 22.29 | 39.23 | 66.89 | 64.63 |
| InCP | In-context | **72.27** | 47.58 | **90.62** | **67.54** | **70.26** | **94.69** | 84.62 | **24.99** | **64.52** | **67.93** | **68.50** |

Table 3: Comparison with existing methods across a range of distribution shifts on the ImageNet dataset. CoOp, CoCoOp, CLIPOOD, MaPLe and PromptSRC are trained on source domain (ImageNet) and evaluated on out-of-domain datasets (ImageNet variant distribution). Baseline CLIP, prompt ensemble, and TPT don't require ImageNet dataset as training data.

| Method | Type | ImageNet Top 1 acc. | ImageNet-A Top 1 acc. | ImageNet-V2 Top 1 acc. | ImageNet-R Top 1 acc. | ImageNet-S Top 1 acc. | Average |
|---|---|---|---|---|---|---|---|
| CLIP-ViT-B/16 | Zero-shot | 66.73 | 47.87 | 60.86 | 73.98 | 46.09 | 59.11 |
| Ensemble (Shu et al., 2022) | Zero-shot | 68.34 | 49.89 | 61.88 | 77.65 | 48.24 | 61.20 |
| TPT (Shu et al., 2022) | Zero-shot | 68.98 | 54.77 | 63.45 | 77.06 | 47.94 | 61.44 |
| CoOp (Zhou et al., 2022b) | Few-shot | 71.51 | 49.71 | 64.20 | 75.21 | 47.99 | 61.72 |
| CoCoOp (Zhou et al., 2022a) | Few-shot | 71.02 | 50.63 | 64.07 | 76.18 | 48.75 | 62.13 |
| CLIPOOD (Shu et al., 2023) | Few-shot | 71.60 | 50.4 | **64.9** | 77.2 | – | – |
| MaPLe (Khattak et al., 2023) | Few-shot | 70.72 | 50.90 | 64.07 | 76.98 | 49.15 | 62.36 |
| PromptSRC (khattak et al., 2023) | Few-shot | 71.27 | 50.90 | 64.35 | **77.8** | **49.55** | 62.77 |
| InCP | In-context | **71.62** | **56.51** | 63.87 | 77.63 | 48.08 | **63.54** |

of plant or animal species (Flowers102 (Nilsback & Zisserman, 2008), OxfordPets (Parkhi et al., 2012)), food (Food101 (Bossard et al., 2014)), transportation (StanfordCars (Krause et al., 2013)), scenes Xiao et al. (2010), aircraft (Aircraft (Maji et al., 2013)), textures (DTD (Cimpoi et al., 2014)), satellite images (EuroSAT (Helber et al., 2019)), human actions (UCF101 (Soomro et al., 2012)), and general objects (Caltech101 (Fei-Fei et al., 2004)). OOD datasets is from ImageNet variants, i.e. ImageNet-V2 (Recht et al., 2019), ImageNet-Sketch (Wang et al., 2019), ImageNet-A (Hendrycks et al., 2021b) and ImageNet-R (Hendrycks et al., 2021a). For more detailed information about the datasets and implementation details, we refer readers to Section A.1 in the appendix.

## 4.1 GENERALIZE CLIP TO FINE-GRAINED CLASSIFICATION DATASETS

In this experiment, we investigate the cross-dataset generalization ability of CLIP model from ImageNet to various fine-grained datasets. Experimental results are reported in table 2. It can be seen that all prompt learning approaches perform better than the zero-shot generalization of CLIP model. However, the learned few-shot prompts (i.e., CoOp (Zhou et al., 2022b), CoCoOp (Zhou et al., 2022a)) are constrained by the distribution corresponding to training data and may exhibit limited generalization beyond that, especially for fine-grained datasets. The learned zero-shot prompts (i.e., TPT (Shu et al., 2022)) catastrophically underfit to novel tasks, whereas our model successfully learns all tasks with the help of task-specific examples. As demonstrated in Table 2, our model significantly outperforms zero-shot and few-shot baselines.

## 4.2 GENERALIZE CLIP TO DISTRIBUTION SHIFT

In this experiment, we evaluate the out-of-distribution generalization of CLIP model on different variants of ImageNet. A particular in-distribution dataset is ImageNet, while we use several out of distribution datasets from ImageNet variants, i.e. ImageNet-V2 (Recht et al., 2019), ImageNet-Sketch (Wang et al., 2019), ImageNet-A (Hendrycks et al., 2021b) and ImageNet-R (Hendrycks et al., 2021a). The performance on four variants of ImageNet with distribution shifts are summaried in Table 3. As shown in Table 3, CoOp (Zhou et al., 2022b), CoCoOp (Zhou et al., 2022a), CLIPOOD (Shu et al., 2023), MaPLe (Khattak et al., 2023) and PromptSRC (khattak et al., 2023) are few-shot prompt tuning methods, which are trained on ImageNet using 16-shot training data per category. This is a type of rapid adaptation from ImageNet to ImageNet Variants, thus achiev-

ing better classification accuracies than zero-shot generalization of CLIP model. TPT (Shu et al., 2022) learn the language prompt using only the given test sample and enhances the zero-shot performance of CLIP model without any task-specific training data. Our InCP extends TPT to visual prompt learning and learn task-specific information from in-context examples of testing data distribution. While our InCP is not trained on ImageNet dataset, it inherits the zero-shot abilities of CLIP model and achieve on-par generalization as ImageNet trained baselines (CoOp (Zhou et al., 2022b), MaPLe (Khattak et al., 2023), PromptSRC (khattak et al., 2023)), surpassing existing methods in terms of average accuracy.

## 4.3 ABLATION STUDIES

**Component Analysis**. In this experiment, we assess the impact of each component within our method on both task and instance adaptation. With only a limited number of in-context examples, task-specific adaptation employs them as training data

Table 4: Study on task and instance-specific adaptations. w/o U/S/SS-obj: without unsupervised/supervised/semi-supervised objective.

| Method | Adaptation | Flower Top 1 acc. | DTD Top 1 acc. | Pets Top 1 acc. |
|---|---|---|---|---|
| Ours w/o U-obj | Task | 69.39 | 42.43 | 87.84 |
| Ours w/o S-obj | Instance | 67.24 | 24.05 | 87.11 |
| Ours (SS-obj) | Task & Instance | **71.13** | **47.34** | **90.60** |

to fine-tune a task-aware prompt, which can then be applied to any test sample from a new task. In contrast, sample-specific adaptation does not depend on in-context examples; instead, it learns a sample-aware prompt using only unlabeled test samples. To investigate the effect of task- and instance-specific adaptation, We create three distinct variants of our method: **(1) Ours without supervised objective (w/ U-obj)** does not adapt task-specific labeled examples for each task in prompt learning; **(2) Ours without unsupervised objective (w/o U-obj)** does not utilize on-the-fly unlabeled test sample to learn sample-specific patterns; **(3) Ours with semi-supervised objective (SS-obj)** adopt in-context examples as domain-specific context information and employ test sample to query model to directly predict the output label.

Results are reported in Table 4. The two methods without supervised/unsupervised objectives show lower performance than ours with semi-supervised objective. This demonstrates that incorporating task-specific and instance-specific adaptation are both beneficial for fine-grained vision recognition problems. Task-specific adaptation is a crucial component of our problem, as instance-specific adaptation is limited to tuning prompt tokens solely on unlabeled samples using unsupervised objectives. Our method learns prompt tokens directly from task-specific image-label pairs and instance-specific test sample to adapt model accordingly.

**Effect on In-context Examples and Unlabeled Test Sample.** In-context examples do not belong to the same categories as the unlabeled test samples. Consequently, they provide task-specific cross-modality alignment information rather than class-specific information, as is typical in traditional few-shot learning. The test samples are unlabeled and only contribute visual modality information to the CLIP model, lacking any language information.

Table 5: Comparison of different prompt learning strategies with in-context examples and one-shot scenarios. V-P: vision-prompt, T-P: Text-prompt, Con-P: Concurrent-prompt, Cyc-P: Cyclic-prompt. In-con: In-context examples.

| Method | Type | Flower Top 1 acc. | DTD Top 1 acc. | Pets Top 1 acc. |
|---|---|---|---|---|
| CLIP-ViT-B/16 | 0-shot | 67.44 | 44.27 | 88.25 |
| InCP w/ V-P | In-Con | 71.13 | 47.34 | 90.60 |
| InCP w/ T-P | In-Con | 69.10 | 45.45 | 88.39 |
| InCP w/ Con-P | In-Con | 69.71 | 46.04 | 88.50 |
| InCP w/ Cyc-P | In-Con | **72.27** | **47.58** | **90.62** |
| InCP w/ V-P | 1-shot | 71.99 | 46.57 | 90.35 |
| InCP w/ T-P | 1-shot | 76.65 | 50.77 | 90.81 |
| InCP w/ Con-P | 1-shot | 76.09 | 51.60 | 92.04 |
| InCP w/ Cyc-P | 1-shot | **82.62** | **53.84** | **95.86** |

As shown in Table 5, visual prompt learning significantly improves the zero-shot performance of CLIP compared to language prompt learning. We also evaluate the result of our InCP when visual and language prompts are concurrently tuned, referred to as "concurrent-prompt", which yields inferior performance compared to single visual prompt learning. Further, we consider an image-label pair from the same categories with test sample and conduct one-shot learning for our approach. The one-shot can provide class-specific cross-modality alignment information for unlabeled test sample. Compared to in-context learning, one-shot learning significantly improves the performance of InCP w/ concurrent-prompt. We argue that concurrent-prompt does not explore the class-specific contextual information of one-shot sample. As shown in Table 5, the proposed cyclic prompt learning achieve the best accuracies and outperform concurrent-prompt with large margin (e.g., +6.53 for

Flower dataset). This can be attributed to the disentanglement of contextual information into different modalities through cyclic learning. Here we perform a single iteration of visual and language prompt with sequential learning, and additional iterations are expected to yield improved results. We leave the exploration of iterative training for future research.

.4**Comparison with Previous Visual Prompt Approaches.** Apart from our proposed language-aware visual prompting, we also evaluate different visual prompting methods, including pixel-level patched, padded prompts (Bahng et al., 2022; Jia et al., 2022)

Table 6: Accuracy of different visual prompt methods.

| Method | Flower Top 1 acc. | DTD Top 1 acc. | Pets Top 1 acc. |
|---|---|---|---|
| Patched prompt (Bahng et al., 2022) | 57.90 | 32.51 | 77.98 |
| Padded prompt (Jia et al., 2022) | 56.07 | 32.68 | 79.39 |
| Token prompt | 59.32 | 34.63 | 74.19 |
| Generic-language prompt | 63.30 | 44.74 | 85.20 |
| Ours (Language-aware) | **71.13** | **47.34** | **90.25** |

and sequence-level token prompt with random initialization. As shown in Figure 6, we learn a single image perturbation by adding learnable patched and padded prompt to input image, which are respectively denoted as "patched prompt" and "padded prompt". Following Jia et al. (2022), we also prepend some learnable parameters into input sequence of transformer, denoted as "token prompt". Table 6 shows the accuracies with different prompt designs. Compared with patched-prompt, padded-prompt, and token-prompt, our proposed visual prompt provides a robust language-aware initialization for model optimization, which not only enhances prompt-based performance but also contributes to increased training stability. To further examine the impact of parameter initialization, we experiment with generic-language prompt, which uses a generic language initialized with random context vectors. As shown in Table 6, generic-language prompt show inferior performance to our language-aware prompt but outperforms the language-unaware prompt methods (i.e., padded, patched and token prompts). This demonstrates that incorporating language modality information into visual branch is beneficial to prompt-based visual recognition.

.5**Does the number of in-context examples matter?** Recall that in-context examples are sampled from the demonstration pool, which contains one sample from each category. We are interested in understanding whether the quantity of these examples impacts performance, particularly for the supervised prompt retrieval method. To investigate this, we varied the number of in-context examples from 1 to 19, resulting in a set of results illustrated in Figure 5. Firstly, it's evident that using in-context examples significantly outperforms the "no examples" method, and the performance clearly benefits

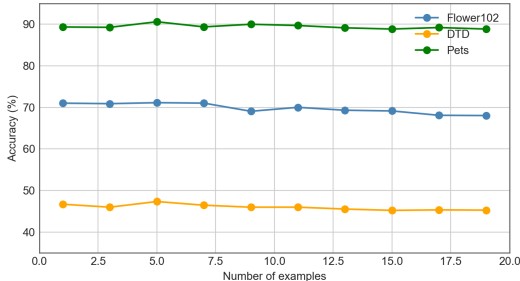

Figure 5: Experimental results w.r.t varied number of in-context examples.

from a larger number of examples. However, interestingly, increasing the number of examples beyond a certain point (i.e., 5) starts to decrease their performance. This phenomenon may be attributed to the fact that in-context examples can be considered as a form of "training data," and an abundance of training data may lead the model to learn prompts specific to the in-context examples rather than the unlabeled test samples. While having more examples may not necessarily be advantageous, the crucial question lies in how to strategically select and employ these examples to harness the full potential of the CLIP model. For a comprehensive exploration and discussion of this topic, please refer to the Section A.2 in the Appendix.

## 5 CONCUSION

In this paper, we tackle the challenge of model generalization in scenarios where the model lacks access to source training data but possesses only a very limited number of samples (e.g., just one sample for each category). Unlike few-shot learning, which utilizes only a small number of samples as training data, we treat these samples as domain-specific contextual information for each test sample. As a result, we introduce Visual In-Context Prompt Learning, a method that empowers a pre-trained vision-language model to make full use of in-context examples. Additionally, we have developed an effective language-aware prompt along with an example selection strategy and implemented a cyclic learning technique to facilitate the seamless integration of the vision prompt with the text prompt. Our experimental results across various downstream tasks consistently demonstrate the effectiveness of our approach.

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

## A APPENDIX

### A.1 DATASETS AND IMPLEMENTATION DETAILS

#### A.1.1 DATASETS

**Datasets.** We assess the transfer performance of our method across 9 fine-grained classification datasets and 4 out-of-distribution (OOD) datasets from ImageNet. Since our method primarily focuses on test-time model adaptation, our evaluation is exclusively based on the testing dataset across all these datasets.

• **Flower102** (Nilsback & Zisserman, 2008) is a widely used dataset consisting of 102 different categories of flowers. Each category consists of between 40 and 258 images. It is commonly employed for fine-grained image classification tasks.

• **OxfordPets** (Parkhi et al., 2012) is a dataset designed for pet image classification, containing a large variations in scale, pose, and lighting conditions. It contains images of 37 different pet breeds with roughly 200 images for each class.

• **Food101** (Bossard et al., 2014) is a dataset specifically curated for food recognition applications. It contains images of 101 different food categories, making it suitable for tasks related to food image classification and analysis.

• **Describable Textures Dataset (DTD)** (Cimpoi et al., 2014) is a texture dataset, which is consists of 5640 images. These images are classified into 47 distinct categories, inspired by human perception, with precisely 120 images allocated to each category.

• **StanfordCars** (Krause et al., 2013) is a dataset commonly used for fine-grained car classification tasks. This dataset contains 16,185 images of 196 classes of cars, which is split into 8,144 training images and 8,041 testing images.

• **Aircraft** (Maji et al., 2013) dataset contains 10,200 images of various aircraft, with 100 images for each of 102 different aircraft model variants, most of which are belonging to airplanes.

• **UCF101** (Soomro et al., 2012) is a widely recognized dataset for human action recognition, which consists of 13,320 video clips spanning 101 different human action categories. These 101 categories are further classified into 5 types, including Body motion, Human-object interactions, Playing musical instruments and Sports, Human-human interactions.

• **EuroSAT** (Helber et al., 2019) is a dataset and deep learning benchmark designed for land use and land cover classification. It is based on Sentinel-2 satellite images with 13 spectral bands and a total of 27,000 labeled and geo-referenced images with 10 distinct classes.

• **Caltech101** (Fei-Fei et al., 2004) dataset is composed of approximately 9,000 images with 101 object categories and a background category. Each object category contains approximately 40 to 800 images, with typical image sizes of 200-300 pixels.

• **SUN397** dataset encompasses 108,753 images spanning 397 categories, serves as a benchmark in scene understanding studies. Each category in this diverse collection is represented by a minimum of 100 images.

• **ImageNet** (Deng et al., 2009) dataset is a large-scale ontology of images built upon the backbone of the WordNet structure, designed to advance the field of computer vision. It spans 1000 distinct object classes and contains 1,281,167 training images, 50,000 validation images and 100,000 test images.

• **ImageNet-V2** (Recht et al., 2019) is a test set including natural images collected from various sources. This dataset is consisted of 10,000 images distributed across 1,000 distinct ImageNet categories.

• **ImageNet-Sketch** (Wang et al., 2019) dataset consists of 50000 images, 50 images for each of the 1000 ImageNet classes. These images are obtained by performing Google Image searches using the query "sketch of [standard class name]."

• **ImageNet-A** (Hendrycks et al., 2021b) is a challenging dataset containing real-world, unmodified, and naturally occurring examples that are misclassified by ResNet models. It includes 7,500 intentionally altered and corrupted images with 1,000 categories to assess the robustness of different models.

• **ImageNet-R** (Hendrycks et al., 2021a) collects images of ImageNet categories presented in artistic renditions, including a total of 30,000 images across 200 distinct ImageNet categories.

### A.1.2 IMPLEMENTATION DETAILS

**Implementation Details.** We apply in-context prompt learning on a pre-trained ViT-B/16 CLIP model. We minimize the semi-parametric objective loss to optimize the visual prompt for 1 step and cyclically alternate between optimizing the visual and prompt for 2 steps. All models are trained with in-context examples and test sample on a single NVIDIA GPU. We use the AdamW (Loshchilov & Hutter, 2017) optimizer with a learning rate of $5e^{-3}$ for all datasets. By default, we set weight parameter $\lambda = 0.4$. If not specifically emphasized, we adopt in-context prompt learning with only visual prompt as our default setting for all ablation studies. We select 5 in-context examples from a fixed subset of labeled data, which is composed by randomly sampling 1 sample from each category. The context samples only provide the task information to do on the test sample. They are usually from the same target dataset, while there is no other relationships between them, e.g., category. Token net is randomly initialized at the start of test-time adaptation and accumulatively updated across the entire evaluation process. For each test sample, $P_t$ is initialized with prefix tokens derived from "a photo of a", which is then converted into a visual token. $P_v$ is then initialized by the above learned $P_t$.

### A.2 ADDITIONAL ABLATION STUDIES

**In-context example selection: random *vs* definition.** This experiment aims to evaluate two distinct approaches for selecting context examples to prompt each test sample: random and definition-based approaches. In the former approach, input-label pairs are randomly selected from candidate examples as context examples for each test sample, while the latter approach utilizes a common set of examples shared across all test samples once the context examples have been sampled. As depicted in Figure 6, the definition-based selection approach also exhibits significant fluctuations with different random seeds, and its performance consistently lags behind the random-based selection approach. The primary reason behind this discrepancy lies in the fact that definition-based examples cannot guarantee the provision of useful context information for all test samples, whereas random-based examples consistently provide each test sample with domain-specific information pertaining to the target distribution.

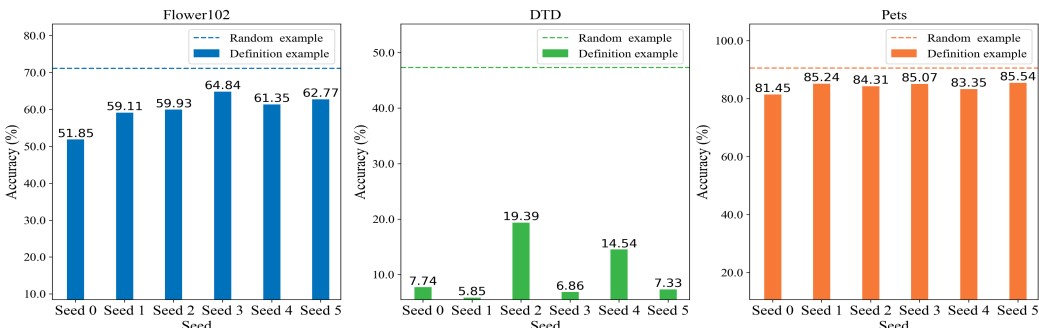

Figure 6: Experimental results w.r.t. different in-context example selection strategies.

**Ground Truth Matters.** To study the impact of correctly-paired inputs and labels in the in-context examples, referred as "ground truth input-label mapping", we evaluate the following three methods.

• **No example** is a typical test-time adaptation method that does not use any labeled data. A prediction is made via optimizing unsupervised objective involving unlabeled test sample.

• **Examples w/o labels** is the baseline that only uses input images without the corresponding labels, which indicates the model performance without looking at the label information.

Table 7: Comparative analysis of inference time and accuracy with existing TPT. The inference time (Infer. Time) is calculated in minutes.

| Method | Flower102 | | Pets | | Cars | | Caltech101 | |
|---|---|---|---|---|---|---|---|---|
| | Infer. Time (↓) | Top 1 acc. (↑) | Infer. Time (↓) | Top 1 acc. (↑) | Infer. Time (↓) | Top 1 acc. (↑) | Infer. Time (↓) | Top 1 acc. (↑) |
| TPT (Shu et al., 2022) | 97.22 | 68.98 | 111.31 | 87.79 | 322.59 | 66.87 | 86.25 | 94.16 |
| InCP (Ours) | **23.79** | **72.27** | **16.31** | **90.62** | **69.05** | **67.54** | **35.54** | **94.69** |

• **Examples w/ gold labels** are employed in a typical in-context learning method with a set of labeled examples, which indicates the model performance with looking at relevant knowledge.

• **Examples w/ random labels** replace all gold labels with random labels, which are randomly sampled at uniform from label space on the testing data.

• **Examples w/ same labels** replace all gold labels with the same label, which is consistent with the label of test sample.

• **Examples w/ oracle labels** replace traditional in-context examples with the oracle examples, where labels are consistent with the label of test sample.

In Fig. 7, we present the recognition accuracy values obtained with different input-

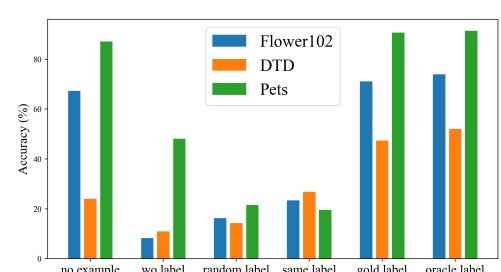

Figure 7: Results when using no-examples, examples w/o labels, examples w/ random labels, examples w/ same labels, examples w/ gold labels and examples w/ oracle labels in fine-grained classification task.

label strategies applied to in-context examples. It is evident that model performance is notably sensitive to the correctness of labels. Specifically, using correct gold labels yields better results than employing random labels. Furthermore, employing the same labels as the test sample leads to a significant improvement compared to using random labels. This observation indicates that having consistent labels provides the model with instructive information about the test samples. Moreover, employing examples with oracle labels achieves the upper bound in performance and outperforms the use of examples with gold labels. We attribute this phenomenon to the increased instructive information, which arises not only from the example labels but also from the example inputs themselves. These results underscore the substantial impact of in-context examples' labels on In-Context Learning (ICL) performance, aligning with the findings (Wu et al.).

**Training strategy: In-context learning *vs* few-shot learning.**

A conventional approach for leveraging the in-context examples dataset involves fine-tuning the prompt on a labeled dataset. As an alternative method, we implemented few-shot prompt learning using all available in-context examples and conducted a comparison with our InCP approach. As demonstrated in Figure 8, the image representations learned through

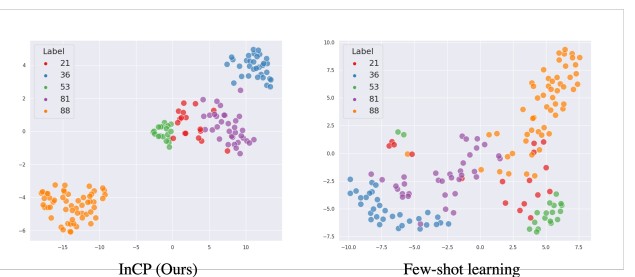

Figure 8: Comparison with few shot learning when using few in-context examples on Flower102.

the few-shot approach exhibit lower resolution in distinguishing between classes compared to our prompted representations. The few-shot approach might introduce more variance and noise into the feature space, and the learned features are specific to the task rather than the individual test sample. In contrast, our InCP method is specifically designed to learn an informative prompt for each test sample using in-context examples, enabling a more effective alignment between the sample and its associated examples.

**Comparative analysis of inference time metrics with existing TPT.** Table 7 reports the comparative analysis of inference time metrics with TPT Shu et al. (2022). All experiments are conducted on one 2080 Ti GPU, and inference time (Infer. Time) is calculated in minutes. The table shows that TPT Shu et al. (2022) needs significant inference time due to augmenting images by 64 times. In

Table 8: Comparison with CoOp and CoCoOp using the same examples.

| Method | Flower102 | DTD | Pets | Cars | Caltech101 |
|---|---|---|---|---|---|
| | Top 1 acc. | Top 1 acc. | Top 1 acc. | Top 1 acc. | Top 1 acc. |
| CoOP (Zhou et al., 2022b) | 66.10 | 30.97 | 82.77 | 60.20 | 90.26 |
| CoCoOP (Zhou et al., 2022a) | 67.23 | 31.72 | 83.14 | 59.78 | 90.43 |
| Ours | **71.13** | **47.34** | **90.60** | **67.54** | **94.69** |

Table 9: Study on task and instance-specific adaptations. w/o U/S/SS-obj represents without unsupervised/supervised/semi-supervised objective.

| Method | Adaptation | Imagenet-R | ImageNet-S | Cars | Caltech101 |
|---|---|---|---|---|---|
| | | Top 1 acc. | Top 1 acc. | Top 1 acc. | Top 1 acc. |
| Ours w/o U-obj | Task | 76.89 | 47.24 | 65.29 | 93.96 |
| Ours w/o S-obj | Instance | 75.73 | 44.51 | 59.08 | 93.10 |
| Ours (SS-obj) | Task & Instance | **77.56** | **48.03** | **67.54** | **94.69** |

contrast, our InCP only uses few in-context examples (i.e., 5) without any augmentation, requiring less inference time.

**Comparison with CoOP and CoCoOP using the same examples.** We provide CoOp/CoCoOp's results using the same examples as InCP in Table 8. Experimental results show that our InCP achieves better performance than CoOp and CoCoOp on the this setting.

**Ablation studies on other datasets.** To provide a more comprehensive evaluation, we perform additional ablation studies on other datasets, including ImageNet-R, ImageNet-S, Cars, and Caltech101. The results are detailed in Table 9 and Table 10.

Table 10: Accuracy of different visual prompt methods.

| Method | Imagenet-R | ImageNet-S | Cars | Caltech101 |
|---|---|---|---|---|
| | Top 1 acc. | Top 1 acc. | Top 1 acc. | Top 1 acc. |
| Patched prompt (Bahng et al., 2022) | 70.62 | 43.69 | 65.14 | 91.68 |
| Padded prompt (Jia et al., 2022) | 68.54 | 40.48 | 55.93 | 89.61 |
| Token prompt | 70.39 | 40.10 | 57.22 | 85.60 |
| Generic-language prompt | 76.46 | 45.29 | 64.17 | 92.01 |
| Ours (Language-aware) | **77.56** | **48.03** | **67.54** | **94.69** |

## A.3 DISCUSSION

In-context learning allows large language models (GPT3 Brown et al. (2020), LaMMa Touvron et al. (2023)) to perform inference on unseen tasks by conditioning on in-context examples (a.k.a. prompt) without updating the model parameters. Inspired by this, existing works Wang et al. (2023b); Zhang et al. (2023); Wang et al. (2023a) explore "in-context learning" concept for vision model, in which the model is updated using in-context examples. Meanwhile, CLIP itself is not able to conduct in-context learning task. To equip CLIP with this ability, our InCP introduces learnable prompt for each test sample in test-time stage. In this way, the model can automatically understand the underlying task with in-context examples.

**Comparison with few-shot learning.** As few-shot methods, CoOP Zhou et al. (2022b) and Co-CoOP Zhou et al. (2022a) fine-tune the prompt on ImageNet dataset using 16-shot training data per category and evaluate the generalization performance on downstream tasks. Our work primarily differs from few-shot methods in two main aspects, i.e., sample selection and quantity. (a) For sample selection, few-shot uses strict categories with specific number of samples, which are widely used in training stage. Differently, in-context learning has no constraint on category. The in-context samples in testing stage can either share the same category as the current test sample or the irrelevant category. It is also impractical to know the exact category of unlabeled test sample in advance. (b)

For sample quantity, few-shot learning requires a predefined number of samples from each category, while in-context learning uses a small, arbitrary set of labeled samples-commonly just five samples.

**Comparison with semi-supervised learning.** Semi-supervised learning Tarvainen & Valpola (2017); Sohn et al. (2020) typically incorporates labeled data during the training phase, amalgamating it with unlabeled data to fine-tune the model and improve its performance on unlabeled samples. Labeled data in semi-supervised learning often shares categories with the unlabeled data. In our method, there is no inherent relationship between in-context examples (labeled data) and the test sample, as they are both drawn from the same domain dataset. Our approach does not necessitate any category information about the test sample, distinguishing it from semi-supervised learning methods.

