# OpenReview forum: "In-context Prompt Learning for Test-time Vision Recognition with Frozen Vision-Language Model"
_ICLR.cc/2024/Conference — Submitted to ICLR 2024_

### Official Review · Reviewer_8csS · 2023-10-23

**Soundness:** 2 fair
**Presentation:** 3 good
**Contribution:** 2 fair
**Rating:** 5
**Confidence:** 3

**Summary:**

This paper adapts visual prompt tuning to TTA of vision-language models. They use a token net and more examples to increase the generalization of vision-language models on domain-specific tasks.

**Strengths:**

The introduction of visual prompt tuning is interesting.

**Weaknesses:**

The pros and cons of the proposed method are not discussed thoroughly.

**Questions:**

- 1. Why use the term "in-context"? What is the "context" for an incoming test sample? What is the relationship between the context samples and the test sample?

- 2. The goal of TTA is to address the domain shift on the fly. The introduction of the context samples makes the model task-specific. For each dataset in Table 2 and Table 3, do we need to construct a dataset-specific context?
    - 2.1 If the answer is YES, I think this is not a TTA method. It is a domain/dataset-specific 1-shot or few-shot approach.
    - 2.2 If the answer is NO, the method is like [TPT with CoOp weights]. It is a one-shot training + test time visual prompt tuning. However, TPT+CoOp has much better performance.
    - 2.3 The authors should further clarify how the context samples are selected. In Table 4, we can see the context samples are vital in the proposed method.

- 3. Please also provide the inference time of the proposed method to evaluate the efficiency.

---

> ### Author Response · Authors · 2023-11-20
>
> Thank you for your detailed review of our paper and valuable feedback! We are encouraged that you found our work interesting and promising. In the following, we will answer your questions one by one.
>
> >Q1: Why use the term "in-context"? What is the "context" for an incoming test sample? What is the relationship between the context samples and the test sample?
>
> A1: i) In-context learning, a new paradigm in NLP, enables large language models to infer unseen tasks by conditioning on specific in-context examples. To harness this emergent ability in large vision-language models like CLIP, we develop a method to learn adaptive visual prompts. This is achieved using just a few image-label pairs, which act as contextual information, aiding the model to swiftly adapt to new test samples. Therefore, these pairs are referred to as in-context examples.
>
> ii) For an incoming test sample, the "context" is in-context samples, i.e., several provided image-label pairs, to let the model do task like these in-context samples do on the test sample.
> iii) The context samples only provide the task information to do on the test sample.
> They are usually from the same target dataset, while there is no other relationships between them, e.g., category.
>
> >Q2: The goal of TTA is to address the domain shift on the fly. The introduction of the context samples makes the model task-specific. For each dataset in Table 2 and Table 3, do we need to construct a dataset-specific context? If the answer is YES, I think this is not a TTA method. It is a domain/dataset-specific 1-shot or few-shot approach.
>
> A2:  Yes. We need to construct a dataset-specific context for each dataset in Table 2 and Table 3.
>
> However, our InCP is not a domain/dataset-specific 1-shot or few-shot approach.
> The reasons are: i) domain-specific few-shot approach is for training process, while our InCP is in the test time; ii) few-shot approach needs at least one sample from each category for training, while in-context examples in our InCP have no this constraint. We only sample five in-context examples no matter what category they belong to. iii) few-shot approaches like CoOP [1]  and CoCoOP [2] employ a shared prompt for all test samples, whereas our InCP adopts an adaptive, sample-specific prompt for each individual test sample.
>
>
> >Q3: The authors should further clarify how the context samples are selected.
>
> A3: We employ only a few number (i.e., 5) of in-context examples. Those examples are randomly chosen from a designated subset of the target dataset, termed "candidate in-context examples". Such a subset is created by selecting one image-label pair from each category, ensuring a diverse and representative sample pool.
>
>
> >Q4: Please also provide the inference time of the proposed method to evaluate the efficiency.
>
> A4:  Thanks for your suggestion.
> We report the inference time of the proposed method in the following table.
>
> | Method                 | Flower102 Infer. Time (↓) | Flower102 Top 1 acc. (↑) | Pets Infer. Time (↓) | Pets Top 1 acc. (↑) | Cars Infer. Time (↓) | Cars Top 1 acc. (↑) | Caltech101 Infer. Time (↓) | Caltech101 Top 1 acc. (↑) |
> | --- | --- | --- | --- | ---| ---| ---| --- | ------------------------- |
> | TPT [1] | 97.22  | 68.98     | 111.31       | 87.79    | 322.59  | 66.87    | 86.25  | 94.16     |
> | InCP (Ours)            | **23.79**                 | **72.27**                | **16.31**            | **90.62**           | **69.05**            | **67.54**           | **35.54**                  | **94.69**                 |
>
>
> [1] Kaiyang Zhou, Jingkang Yang, Chen Change Loy, and Ziwei Liu. Learning to prompt for vision-language models. In IJCV, 2022.
>
> [2] Kaiyang Zhou, Jingkang Yang, Chen Change Loy, and Ziwei Liu. Conditional prompt learning for vision-language models. In CVPR, 2022.

---

> > ### Comment · Reviewer_8csS · 2023-11-23
> > **Response**
> >
> > Thanks for the author's review.
> >
> > Well, I think the dataset-specific context selection process will largely limit the application of the proposed methods. The creation of the context put a strong prior on the method. Further discussion and justification are needed.
> >
> > So I maintain my previous score, i.e., marginally below.

---

> ### Author Response · Authors · 2023-11-23
>
> Thank you for your feedback! We provide responses to your comments below:
>
> >Q1: I think the dataset-specific context selection process will largely limit the application of the proposed methods. The creation of the context put a strong prior on the method. Further discussion and justification are needed.
>
> A1: While our method is originally tailored for specific datasets, its adaptation to large-scale datasets such as ImageNet allows the extraction of significant context information. This context information serves as an in-context prompt that can be applied to different datasets or downstream tasks. The versatility introduced through this approach enables our method to generalize across various datasets, thereby enhancing its overall applicability.

---

### Official Review · Reviewer_QS1Q · 2023-10-23

**Soundness:** 3 good
**Presentation:** 2 fair
**Contribution:** 3 good
**Rating:** 5
**Confidence:** 5

**Summary:**

This paper focused on the task of test-time prompt learning for vision-language models like CLIP, and proposed a visual in-context learning method to conduct one-time prompt tuning using both labeled samples as context and unlabeled test sample in a semi-supervised learning manner. The method updates both visual prompt and language prompt. Experiments are conducted on fine-grained classification datasets and distribution shift setting.

---

After rebuttal: My concerns on the comparison with few-shot learning and the information leakage of in-context samples remain. The authors confirmed that the in-context samples are guaranteed not to share the same label as the test sample, and the model can benefit from avoiding predicting the categories of in-context samples. In other words, the in-context samples help to filter out wrong answers and improve the performance, which is unfair for evaluation.

**Strengths:**

+ The task of test-time prompt tuning is relatively new and worth investigating. The idea of in-context prompt learning for vision-language models is novel to me.

+ The proposed method achieved consistent improvement on fine-grained classification datasets, and competitive performance on distribution shifting datasets.

**Weaknesses:**

- (Major concern) This work proposed a in-context prompt learning method that used labeled samples as context during test stage. However, different from traditional in-context learning that used additional samples as a part of prompts and did not tune any parameters, these method used additional samples to tune the network rather than serve as part of prompt, which is like traditional semi-supervised learning pipeline but with fewer data and training loops. I am wondering whether the terminology of "in-context prompt learning" is proper. More evidence or reference on the terminology definition is expected during discussion.

- (Major concern) Random sampling may cause the information leakage of test set. Sec. 3.1 and Appendix A.2 indicate that the labeled data are randomly sampled for different test samples, which means that the model may see a wide range of test samples during evaluation, although they are not simultaneously seen. Compared to few-shot tuning setting, only a fixed set of labeled data is accessed. As shown in Appendix A.2, a fixed set of labeled data underperforms random sampling. I am wondering whether the performance gain is due to the information leakage caused by random sampling. A strict implementation would be using sampling from the same subset of labeled data for rather than the whole test set.

- The figures are confusing and not well designed. First, in Figure 2, it is not clear what does token space mean. The number of input tokens are 4 and the number of output tokens are 2. Does it mean the number of input tokens equals to the number of words in the prompt, and the token net aims to compress the tokens? Second, in Figure 3, the module and tokens are denoted in many colors, but it is hard to understand what each color indicates, and what is the difference between different tokens in (c) and (d). Third, in Figure, the dimension of $P_t$ (i.e., 3) does not equal to the number of prefix tokens (i.e., 4). Is it in intention? Figure 1(c) is also confusing to me to understand what happened during the cyclic learning stage.

- Difference between this method and fully test-time adaptation. According to Table 1, the main difference is that this method used additional labeled data for tuning the prompts. If so, the novelty is just additional annotations in a semi-supervised manner, which is limited.

- The ablation studies in Tables 4 and 6 are not comprehensive, which are only on three datasets compared to the main results on Table 2. Why only these three datasets are selection? How are they representative?

**Questions:**

Please see Weaknesses for detailed comments.

---

> ### Author Response · Authors · 2023-11-20
>
> Thank you for your detailed review of our paper and constructive feedback! We are encouraged that you found our task promising and our idea novel. We have revised our paper and addressed your concerns. Below, we detail each change made in response to your feedback.
>
>
> >Q1: I am wondering whether the terminology of "in-context prompt learning" is proper. More evidence or reference on the terminology definition is expected during discussion.
>
> A1:  We follow existing works [1,2,3] for this ``in-context learning'' concept, in which the model is updated using in-context examples.
> Meanwhile, CLIP itself is not able to conduct in-context learning task.
> To equip CLIP with this ability, our InCP introduces learnable prompt for each test sample in test-time stage.
> In this way, the model can automatically understand the underlying task with in-context examples. For sample selection, in-context learning has no constraint on category in testing stage. The in-context samples can either share the same category as the current test sample or the irrelevant category. For sample quantity, in-context learning uses a small, arbitrary set of labeled samples. Therefore, our InCP belongs to the in-context learning area.
>
> >Q2: I am wondering whether the performance gain is due to the information leakage caused by random sampling. A strict implementation would be using sampling from the same subset of labeled data for rather than the whole test set.
>
> A2: There is no information leakage in our InCP.
> Actually, the selected 5 in-context examples do not share the same label with the test sample.
> These 5 in-context examples are selected from a fixed subset of labeled data, which is composed by randomly sampling 1 sample from each category.
> For each test sample, we use the same subset rather than the whole test set, which is the same setting as you suggested.
>
> >Q3: The figures are confusing and not well designed. First, in Figure 2, it is not clear what does token space mean. The number of input tokens are 4 and the number of output tokens are 2. Does it mean the number of input tokens equals to the number of words in the prompt, and the token net aims to compress the tokens? Second, in Figure 3, the module and tokens are denoted in many colors, but it is hard to understand what each color indicates, and what is the dfference between different tokens in (c) and (d). Third, in Figure, the dimension of (i.e., 3) does not equal to the number of prefix tokens (i.e., 4). Is it in intention? Figure 1(c) is also confusing to me to understand what happened during the cyclic learning stage.
>
> A3: i) "token space" in Figure 2 refers to the embedding space of text tokens and visual tokens. The number of input tokens equals to the number of words in the prompt and the output tokens.
> Here, the 4 to 2 means the dimension decrease, not the token number. Sorry for the confusion of token number.
> We modify Figure 2 in the manuscript for clarification.
> ii) The orange token represents a learnable visual token, the light purple tokens represent static visual tokens, the light gray tokens represent static text tokens, and the light orange tokens represent static prefix tokens in both Figure 3 (c) and (d). We update Figure 3 in the manuscript.
> iii) Sorry for the confusion. The number of output tokens should be the same as the prefix tokens (i.e., 4). We modify it in the manuscript. iv) During the cyclic learning stage, we first optimize text prompt and then visual prompt in a sequential process. We update Figure 1 (c) for clarification.
>
>
> >Q4: Difference between this method and fully test-time adaptation. According to Table 1, the main difference is that this method used additional labeled data for tuning the prompts. If so, the novelty is just additional annotations in a semi-supervised manner, which is limited.
>
> A4: Fully test-time adaptation is a self-supervised learning method, which applies its self-supervised loss on unlabeled test sample to achieve the generalization capability of model on unseen domains. We argue that self-supervised learning, without task-specific supervision or model modifications, is insufficient for test-time adaptation, and thus propose in-context prompt learning for model adaptation.
>
> Semi-supervised learning typically incorporates labeled data during the training phase, amalgamating it with unlabeled data to fine-tune the model and improve its performance on unlabeled samples. Labeled data in semi-supervised learning often shares categories with the unlabeled data. In our method, there is no inherent relationship between in-context examples (labeled data) and the test sample, as they are both drawn from the same domain dataset. Our approach does not necessitate any category information about the test sample, distinguishing it from semi-supervised learning methods.

---

> > ### Author Response · Authors · 2023-11-20
> >
> > >Q5: The ablation studies in Tables 4 and 6 are not comprehensive, which are only on three datasets compared to the main results on Table 2. Why only these three datasets are selection? How are they representative?
> >
> > A5: We use these three datasets, i.e., Flower, DTD, Pets, since they are reported in most previous methods [4,5,6] and represent various object classification tasks.
> > To provide a more comprehensive evaluation, we perform additional ablation studies on other datasets, including ImageNet-R, ImageNet-S, Cars, and Caltech101. The results are detailed in the below tables.
> >
> > | Method (Top 1 acc.) | Adaptation | Imagenet-R | ImageNet-S | Cars  | Caltech |
> > |:--------|:------------:|:-----------------------:|:-----------------------:|:-----------------:|:-----------------------:|
> > | Ours w/o U-obj | Task | 76.89 | 47.24 | 65.29 | 93.96 |
> > | Ours w/o S-obj | Instance | 75.73 | 44.51 | 59.08 | 93.1 |
> > | Ours (SS-obj) | Task & Instance | **77.56** | **48.03** | **67.54** | **94.69** |
> >
> >
> > | Method (Top 1 acc.)| Imagenet-R | ImageNet-S | Cars | Caltech |
> > |:--------|:--------:|:--------:|:--------:|:--------:|
> > | Patched prompt | 70.62 | 43.69 | 65.14 | 91.68 |
> > | Padded prompt | 68.54 | 40.48 | 55.93 | 89.61 |
> > | Token prompt | 70.39 | 40.10 | 57.22 | 85.60 |
> > | Generic-language prompt | 76.46 | 45.29 | 64.17 | 92.01 |
> > | Ours (Language-aware) | **77.56** | **48.03** | **67.54** | **94.69** |
> >
> > [1] Zhendong Wang, Yifan Jiang, Yadong Lu, Yelong Shen, Pengcheng He, Weizhu Chen, Zhangyang Wang, and Mingyuan Zhou. In-context learning unlocked for diffusion models.
> > arXiv preprint arXiv:2305.01115, 2023.
> >
> > [2] Yuanhan Zhang, Kaiyang Zhou, and Ziwei Liu. What makes good examples for visual in-context learning? arXiv preprint arXiv:2301.13670, 2023
> >
> > [3] Xinlong Wang, Wen Wang, Yue Cao, Chunhua Shen, and Tiejun Huang. Images speak
> > in images: A generalist painter for in-context visual learning. In CVPR, 2023.
> >
> > [4] Manli Shu, Weili Nie, De-An Huang, Zhiding Yu, Tom Goldstein, Anima Anandkumar, and Chaowei Xiao. Test-time prompt tuning for zero-shot generalization in vision-language models. In NeurIPS, 2022.
> >
> > [5] Kaiyang Zhou, Jingkang Yang, Chen Change Loy, and Ziwei Liu. Learning to prompt for vision-language models. In IJCV, 2022
> >
> > [6] Kaiyang Zhou, Jingkang Yang, Chen Change Loy, and Ziwei Liu. Conditional prompt learning for vision-language models. In CVPR, 2022.

---

> > > ### Comment · Reviewer_QS1Q · 2023-11-22
> > >
> > > Thank the authors for the feedback. However, the rebuttal didn't address my concerns well.
> > >
> > > First, the authors claim that in-context learning means `the model is updated using in-context examples.` I share the same concern with Reviewer k8E2 that it is more like few-shot learning. I remain concerned about the terminology and whether the comparisons with previous prompt tuning methods are fair.
> > >
> > > Second, the authors claim that `the selected 5 in-context examples do not share the same label with the test sample`. Does that mean the model can simply avoid predicting these five categories as they are confirmed not the label of the test sample?

---

> > > > ### Author Response · Authors · 2023-11-23
> > > >
> > > > Thank you for your feedback！We provide point-to-point responses to your comments below:
> > > >
> > > > <Q1: It is more like few-shot learning. I remain concerned about the terminology and whether the comparisons with previous prompt tuning methods are fair.
> > > >
> > > > A1: Sorry for the terminology's confusion. We would like to clarify that our method is **not** few-shot learning.
> > > > We understand that the confusion exists because we use in-context samples during the test-time recognition, however, it is different from few-shot learning in two factors.
> > > > i) few-shot learning aims to improve the model's ability during  training, while our InCP is for evaluation stage.
> > > > ii) few-shot learning uses several samples in each category, which share the same categories with the test samples.
> > > > Differently, the categories of in-context samples in our InCP are totally different from the test sample.
> > > >
> > > > Therefore, we classify our method as "in-context" learning.
> > > > This is also motivated by existing works [1,2,3], in which the model is updated using in-context examples.
> > > >
> > > >
> > > > The comparisons with previous prompt tuning methods are fair.
> > > > Actually, existing prompt tuning methods (e.g., CoOp and CoCoOp) are fine-tuned on 16-shot ImageNet training data per category, which the number of training samples is larger than ours.
> > > > TPT and our InCP do not require training data.
> > > > Our InCP is built upon TPT, further using in-context samples (without same label as the test sample) for test-time improvement.
> > > > Thus, the comparisons are fair and our InCP shows better performance.
> > > >
> > > > <Q2 The authors claim that the selected 5 in-context examples do not share the same label with the test sample. Does that mean the model can simply avoid predicting these five categories as they are confirmed not the label of the test sample?
> > > >
> > > > A2: Yes. Our model can avoid predicting examples' categories, and we have implemented "Examples w/o labels" baseline that only uses input images without the corresponding labels, which indicates the model performance without looking at the label information. Experiment results are reported in Figure 7 of main paper. As shown in Figure 7, using labels yields better results than no labels. This indicates that having labels provides the model with instructive information about the test samples.
> > > >
> > > > [1] Yuanhan Zhang, Kaiyang Zhou, and Ziwei Liu. What makes good examples for visual in-context learning? arXiv preprint arXiv:2301.13670, 2023
> > > >
> > > > [2] Xinlong Wang, Wen Wang, Yue Cao, Chunhua Shen, and Tiejun Huang. Images speak
> > > > in images: A generalist painter for in-context visual learning. In CVPR, 2023.
> > > >
> > > > [3] Zhendong Wang, Yifan Jiang, Yadong Lu, Yelong Shen, Pengcheng He, Weizhu Chen, Zhangyang Wang, and Mingyuan Zhou. In-context learning unlocked for diffusion models.
> > > > arXiv preprint arXiv:2305.01115, 2023.

---

### Official Review · Reviewer_k8E2 · 2023-10-29

**Soundness:** 3 good
**Presentation:** 2 fair
**Contribution:** 3 good
**Rating:** 5
**Confidence:** 4

**Summary:**

This work proposes a prompt learning method for vision-language models. It proposed to optimize both textual and visual prompts jointly (via cyclic learning) using an unsupervised objective on the test sample along with a supervised objective on some few-shot (in-context) examples. Experiment results show that it performs better than existing single-modality prompt learning methods.

**Strengths:**

1. The proposed prompt learning method connects textual and visual prompts, which sounds well-motivated. Experiment results also support the idea.
2. The ablation studies are well-designed and comprehensive.

**Weaknesses:**

1. The method and experiment sections need more elaboration. Currently, there are many unclear points about the method itself and experiment set up. The implementation detail in Appendix A.1.2 doesn’t cover all the details. (See my questions below.)
2. I don’t think it is appropriate to call the proposed method an “in-context learning” method. In-context learning in both the language and vision literature does not involve optimizing any parameters including exterior ones like prompts. I think the method proposed in this work is more like few-shot learning, which updates parameters on some example images and then uses the updated parameters to run inference on the test sample.
3. It needs to provide more details about how Token Net is trained. If it is randomly initialized and only optimized for one step at test time, I don’t think it can “translate” text tokens into visual prompts that the vision encoder comprehends.
4. If I understand correctly, the InCP in Section 4.1 is learned using in-context examples from the target dataset. However, CoOp and CoCoOp are learned on the ImageNet dataset. A fair comparison would be to report CoOp/CoCoOp’s results using the same few-shot examples as InCP.

**Questions:**

1. In Figure 4, it seems like both $P_v$ and $P_t$ and the token net are trainable. But the learning objective in Eq (2) only involves $P_v$?
2. Section 3.4 can be more elaborated. Currently, it’s not very clear. For example, it seems like cyclic prompt learning involves multi-step optimization, while in the paragraph above section 3.4, it says the method is a single-step optimization.
3. During testing, when a new test sample comes in, do you reset $P_v$, $P_t$, and TokenNet back to its initial state? Or are the parameter updates accumulated?
4. Is the Token net separately trained before test-time adaptation or randomly initialized? How are $P_v$ and $P_t$ initialized?
5. How many tokens do you use for $P_v$ and $P_t$? How big is the TokenNet?
6. How many in-context examples do you use?
7. I don’t find the “generic-language prompt” baseline in Table 6.

---

> ### Author Response · Authors · 2023-11-20
>
> Thank you for your detailed review of our paper and constructive feedback! We are encouraged that you found our method well-motivated, our experiment comprehensive, our ablation studies well-designed clear. We are addressing each of your comments and questions below:
>
> >Q1: I don’t think it is appropriate to call the proposed method an "in-context learning" method. In-context learning in both the language and vision literature does not involve optimizing any parameters including exterior ones like prompts. I think the method proposed in this work is more like few-shot learning, which updates parameters on some example images and then uses the updated parameters to run inference on the test sample.
>
> A1: i) We follow existing works [1,2,3] for this "in-context learning" concept, in which the model is updated using in-context examples.
> Meanwhile, CLIP itself is not able to conduct in-context learning task. To equip CLIP with this ability, our InCP introduces learnable prompt for each test sample in test-time stage.
> In this way, the model can automatically understand the underlying task with in-context examples.
>
> ii) Our work is different from few-shot learning.
> The main difference lies in two factors, i.e., sample selection and quantity.
> (a) For sample selection, few-shot uses strict categories with specific number of samples, which are widely used in training stage.
> Differently, in-context learning has no constraint on category.
> The in-context samples in testing stage can either share the same category as the current test sample or the irrelevant category. It is also impractical to know the exact category of unlabeled test sample in advance.
> (b) For sample quantity, few-shot learning requires a predefined number of samples from each category, while in-context learning uses a small, arbitrary set of labeled samples-commonly just five samples. Therefore, our InCP belongs to the in-context learning area, rather than the few-shot learning.
>
>
> >Q2: It needs to provide more details about how Token Net is trained. If it is randomly initialized and only optimized for one step at test time, I don’t think it can "translate" text tokens into visual prompts that the vision encoder comprehends.
>
> A2: Yes, the token net is randomly initialized.
> The feasibility of token net is based on CLIP, where the textual and visual encoders are already aligned well in the embedding space. The token net is initialized by text tokens, and then transferred to visual tokens with a few optimization steps on visual examples, which is easy to be converged.
>
>
> >Q3: If I understand correctly, the InCP in Section 4.1 is learned using in-context examples from the target dataset. However, CoOp and CoCoOp are learned on the ImageNet dataset. A fair comparison would be to report CoOp/CoCoOp’s results using the same few-shot examples as InCP.
>
>
> A3: Thanks for your advice. We provide CoOp/CoCoOp’s results using the same examples as InCP in the following table.
> It shows that our InCP achieves better performance than CoOp and CoCoOp on this setting. Following your suggestion, we add this table in the manuscript for fair comparison.
>
> | Method (Top 1 acc.) | Flower | DTD   | Pets  | Cars  | Caltech |
> | ------ | :------: | :-----: | :-----: | :-----: | :----------: |
> | CoOP [2]  | 66.10   | 30.97 | 82.77 | 60.20  | 90.26      |
> | CoCoOP [3] | 67.23  | 31.72 | 83.14 | 59.78 | 90.43      |
> | InCP (Ours)    | **71.13** | **47.34** | **90.6**  | **67.54** | **94.69** |
>
>
> >Q4: In Figure 4, it seems like both $P_{\text{v}}$ and $P_{\text{t}}$ and the token net are trainable. But the learning objective in Eq (2) only involves $P_{\text{v}}$?
>
> A4: Thanks for pointing out this. The set of $P_{\text{v}}$, $P_{\text{t}}$ and token net are all trainable in our method. The completed objective function is presented below. With step is 1 (i.e., $s=1$) for visual prompt learning, the objective function is $$\mathcal{L}(x_t,\mathcal{D},P_{\text{v}},P_{t},\theta)=
> \underset{P_\text{v}, \theta}{\mathrm{argmin}} \{L(x_t, P_\text{v}) + \sum_{(x_i,y_i) \in \mathcal{D}} \lambda L(x_i, y_i, P_\text{v}, P_\text{t}, \theta)\}.$$ With step is 2 (i.e., $s=2$) for text prompt learning, the objective function is $$\mathcal{L}(x_t,\mathcal{D},P_{\text{v}},P_{\text{t}},\theta)=\underset{P_\text{t}}{\mathrm{argmin}} \{L(x_t, P_\text{t}) + \sum_{(x_i,y_i) \in \mathcal{D}} \lambda L(x_i, y_i, P_\text{v}, P_\text{t}, \theta)\},$$
> where $x_t$ is unlabeled test sample, $\mathcal{D}$ contains in-context examples, $\theta$ is the parameter of token net, and $s=1,2$ are the step numbers of model optimization for visual and text prompt, respectively.

---

> > ### Author Response · Authors · 2023-11-20
> >
> > >Q5: Section 3.4 can be more elaborated. Currently, it’s not very clear. For example, it seems like cyclic prompt learning involves multi-step optimization, while in the paragraph above section 3.4, it says the method is a single-step optimization.
> >
> > A5: Thanks for your advice. We carefully update Section 3.4 in the manuscript to make it clear including the following clarification of the cyclic prompt learning.
> >
> > The paragraph above Section 3.4 is focused on visual prompt learning, which is a single-step optimization process.
> > In Section 3.4, we introduce text prompts to integrate visual prompts.
> > This process is a two-step optimization process, where each type of prompt is optimized in sequence, i.e., first visual then textual prompts. It forms the "cyclic prompt learning" concept.
> >
> > >Q6: During testing, when a new test sample comes in, do you reset $P_{\text{v}}$ and $P_{\text{t}}$, and TokenNet back to its initial state? Or are the parameter updates accumulated?
> >
> > A6: When a new test sample comes in, $P_{\text{v}}$ and $P_{\text{t}}$ is reset, while the parameters of the token net are accumulatively updated.
> >
> > >Q7: Is the Token net separately trained before test-time adaptation or randomly initialized? How are $P_{\text{v}}$ and $P_{\text{t}}$ initialized?
> >
> > A7: Token net is randomly initialized at the start of test-time adaptation and accumulatively updated across the entire evaluation process.
> > $P_{\text{t}}$ is initialized with prefix tokens derived from "a photo of a", which is then converted into a visual token.
> > $P_{\text{v}}$ is then initialized by the above learned $P_{\text{t}}$.
> >
> > >Q8: How many tokens do you use for $P_{\text{v}}$ and $P_{\text{t}}$? How big is the TokenNet?
> >
> > A8: The token numbers of $P_{\text{v}}$ and $P_{\text{t}}$ are 4. The token net consists of a single fully connected layer, with 512 and 768 feature dimension for input and output, respectively.
> >
> > >Q9: How many in-context examples do you use?
> >
> > A9: We use a total of 5 in-context examples.
> >
> > >Q10: I don’t find the "generic-language prompt" baseline in Table 6.
> >
> > A10: Sorry for the missing results. We update Table 6 in the manuscript as following.
> >
> > | Method (Top 1 acc.)| Flower | DTD | Pets |
> > |---|:---:|:---:|:---:|
> > | Patched prompt [4] | 57.90 | 32.51 | 77.98 |
> > | Padded prompt [5] | 56.07 | 32.68 | 79.39 |
> > | Token prompt | 59.32 | 34.63 | 74.19 |
> > | Generic-language prompt | 63.30 | 44.74 | 85.20 |
> > | Ours (Language-aware) | **71.13** | **47.34** | **90.25** |
> >
> >
> > [1] Zhendong Wang, Yifan Jiang, Yadong Lu, Yelong Shen, Pengcheng He, Weizhu Chen, Zhangyang Wang, and Mingyuan Zhou. In-context learning unlocked for diffusion models.
> > arXiv preprint arXiv:2305.01115, 2023.
> >
> > [2] Yuanhan Zhang, Kaiyang Zhou, and Ziwei Liu. What makes good examples for visual in-context learning? arXiv preprint arXiv:2301.13670, 2023
> >
> > [3] Xinlong Wang, Wen Wang, Yue Cao, Chunhua Shen, and Tiejun Huang. Images speak
> > in images: A generalist painter for in-context visual learning. In CVPR, 2023.
> >
> > [4] Hyojin Bahng, Ali Jahanian, Swami Sankaranarayanan, and Phillip Isola. Exploring visual
> > prompts for adapting large-scale models. arXiv preprint arXiv:2203.17274, 2022.
> >
> > [5] Menglin Jia, Luming Tang, Bor-Chun Chen, Claire Cardie, Serge Belongie, Bharath Hariharan, and Ser-Nam Lim. Visual prompt tuning. In ECCV, 2022.

---

### Official Review · Reviewer_3epP · 2023-10-31

**Soundness:** 2 fair
**Presentation:** 2 fair
**Contribution:** 2 fair
**Rating:** 5
**Confidence:** 2

**Summary:**

This paper introduces a method called "visual In-Context Prompt learning (InCP)" to enhance the CLIP model's performance in various downstream tasks. InCP enables a pre-trained vision-language model to adapt to new tasks by leveraging in-context examples without changing its core parameters. The key contributions of the paper include the introduction of InCP as an effective method for incorporating in-context information, exploration of language descriptions for visual prompt initialization, and achieving state-of-the-art results across diverse downstream datasets.

**Strengths:**

+ The paper presents a new method by combining natural language processing and computer vision insights.
+ The paper maintains high quality through a well-structured methodology and rigorous experiments, ensuring robustness and reliability in its findings.
+ The paper is exceptionally clear, providing a strong background and conveying complex concepts effectively, making it accessible to a wide audience.

**Weaknesses:**

- Efficiency is a crucial factor in test-time adaptation, given the significance of swiftly adapting to new environments. It is imperative that the paper includes explicit reporting and a comparative analysis of inference time metrics for InCP, especially when compared to existing methods like TPT.
- This paper requires clarification regarding InCP's performance on the SUN397 dataset (utilized in TPT) in Table 1 and the ImageNet dataset (also used in TPT) in Table 2. Providing a comprehensive comparison of InCP's performance on these specific datasets will significantly enhance the paper's clarity and the reader's understanding.
- The performance improvement in Table 2 appears to be marginal. Further explanation or additional results may be needed to demonstrate the significance of the improvement.

**Questions:**

Please refer to the weaknesses.

---

> ### Author Response · Authors · 2023-11-20
>
> Thank you for your comprehensive review, valuable comments, and constructive feedback. We are encouraged that you found our idea insightful, our method well-structured, our experiments rigorous, and our work exceptionally clear. We provide point-to-point responses to your comments below:
>
> >Q1: Efficiency is a crucial factor in test-time adaptation, given the significance of swiftly adapting to new environments. It is imperative that the paper includes explicit reporting and a comparative analysis of inference time metrics for InCP, especially when compared to existing methods like TPT.
>
> A1: The inference time is shown in the following table. All experiments are conducted on one 2080 Ti GPU, and inference time (Infer. Time) is calculated in minutes. The table shows that needs significant inference time due to augmenting images by 64 times. In contrast, our InCP only uses few in-context examples (i.e., 5) without any augmentation, requiring less inference time. This table clearly shows the efficiency of our method.
>
> | Method                 | Flower102 Infer. Time (↓) | Flower102 Top 1 acc. (↑) | Pets Infer. Time (↓) | Pets Top 1 acc. (↑) | Cars Infer. Time (↓) | Cars Top 1 acc. (↑) | Caltech101 Infer. Time (↓) | Caltech101 Top 1 acc. (↑) |
> | --- | --- | --- | --- | ---| ---| ---| --- | ------------------------- |
> | TPT [1] | 97.22  | 68.98     | 111.31       | 87.79    | 322.59  | 66.87    | 86.25  | 94.16     |
> | InCP (Ours)            | **23.79**                 | **72.27**                | **16.31**            | **90.62**           | **69.05**            | **67.54**           | **35.54**                  | **94.69**                 |
>
> >Q2: This paper requires clarification regarding InCP's performance on the SUN397 dataset (utilized in TPT) in Table 1 and the ImageNet dataset (also used in TPT) in Table 2. Providing a comprehensive comparison of InCP's performance on these specific datasets will significantly enhance the paper's clarity and the reader's understanding.
>
> A2: The performance on ImageNet and SUN397 datasets is shown in the following table.
> It shows that our InCP achieves best performance compared with previous zero/few-shot methods.
> In particular, we only use 5 samples yet is better than few-shot methods [2,3].
>
> | Method        | Type       | ImageNet  | SUN397    |
> | ------------- | :----------: | :---------: | :---------: |
> | CLIP-ViT-B/16 | Zero-shot  | 66.73     | 62.59     |
> | Ensemble [1]     | Zero-shot  | 68.34     | 65.63     |
> | TPT [1]          | Zero-shot  | 68.98     | 65.50      |
> | CoOP [2]         | Few-shot   | 71.51     | 64.15     |
> | CoCoOP [3]        | Few-shot   | 71.02     | 66.89     |
> | MaPLe [4]        | Few-shot   | 70.72     | 67.01     |
> | PromptSRC [5]    | Few-shot   | 71.27     | 67.10     |
> | InCP (Ours)         | In-context | **71.62** | **67.93** |
>
>
> >Q3: The performance improvement in Table 2 appears to be marginal. Further explanation or additional results may be needed to demonstrate the significance of the improvement.
>
> A3: As few-shot methods, CoOP [2] and CoCoOP [3] fine-tune the prompt on ImageNet dataset using 16-shot training data per category and evaluate the generalization performance on downstream tasks. As a self-supervised method, TPT is not trained on ImageNet. It utilizes the self-supervised loss with unlabeled test samples to achieve its zero-shot generalization ability. Our InCP follows TPT, while only using a few in-context examples as domain-specific context information. Table 2 in the main paper shows that our method outperforms TPT by notable margins on two datasets (i.e., 72.27\% vs. 68.98\% on Flower dataset, and 70.26\% vs. 68.04\% on UCF101 dataset). Our InCP can also consistently outperforms CoOP and CoCoOP when using unlabeled ImageNet data (i.e., 70.26\% vs. 68.44\% on UCF101 dataset), and using the same in-context examples (i.e., 71.13\% vs. 67.23\% on Flower dataset). The following table shows the comparison with CoOp and CoCoOp using the same in-context examples.
>
> | Method | Flower | DTD   | Pets  | Cars  | Caltech |
> | ------ | :------: | :-----: | :-----: | :-----: | :----------: |
> | CoOP [2]  | 66.10   | 30.97 | 82.77 | 60.20  | 90.26      |
> | CoCoOP [3] | 67.23  | 31.72 | 83.14 | 59.78 | 90.43      |
> | InCP (Ours)    | **71.13** | **47.34** | **90.6**  | **67.54** | **94.69** |
>
> Referece:
>
> [1] Manli Shu, et al. Test-time prompt tuning for zero-shot generalization in vision-language models. In NeurIPS, 2022.
>
> [2] Kaiyang Zhou, et al. Learning to prompt for vision-language models. In IJCV, 2022
>
> [3] Kaiyang Zhou, et al. Conditional prompt learning for vision-language models. In CVPR, 2022.
>
> [4] Muhammad Uzair Khattak, et al. Maple: Multi-modal prompt learning. In CVPR, 2023.
>
> [5] Muhammad Uzair khattak, et al. Self-regulating prompts: Foundational model adaptation without forgetting. In ICCV, 2023.

---

### Meta-Review · Area_Chair_V4F4 · 2023-12-09

**Metareview:**

Four knowledgeable reviewers reviewed this submission. Their initial concerns included (1) the missing inference time analysis which is relevant for TTA methods (3epP, 8csS), (2) the "in-context" terminology used which appeared non-standard (k8E2, QS1Q, 8csS), (3) the experimental evidence which raised concerns on the fairness of comparisons (k8E2), (4) the potential information leakage of the test set (QS1Q), and (5) the positioning of the contribution w.r.t. fully test-time adaption which appeared unclear. The rebuttal partially addressed the reviewers' concerns by e.g. arguing that the used terminology follows the one in a few recent papers, performing an inference time analysis and contrasting the inference time of the method with the baselines, commenting on the fairness of the existing comparisons, and contrasting the proposed approach with fully test-time adaptation. The authors summarized for the AC how they have addressed the reviewers concerns, and in particular the confusion around the "in-context" terminology. During discussion, the reviewers appreciate the clarification of many technical details during rebuttal but remain hesitant. In particular, they find the experimental evidence still unconvincing, they argue that the "in-context" terminology introduced in the referenced papers is presented for very different setups / tasks (e.g. inpainting), and overall they conclude that the significance of the proposed approach requires further justification. The AC agrees with the reviewers that the terminology used in the paper is indeed confusing, as in-context learning - as originally introduced in the context of LLMs - does not require the optimization of any parameters (as acknowledged by the authors as well). The paper would overall require a major revision to properly position the work (e.g. "in-context" vs few-shot), to address the remaining concerns of the reviewers, and to improve the presentation by focusing on reducing the confusion introduced by the terminology. Therefore, the AC recommends to reject and encourages the author to consider the feedback provided by the reviewers to improve future iterations of their work.

**Justification For Why Not Higher Score:**

The paper needs a major revision to properly position the work (e.g. "in-context" vs few-shot), to address the remaining concerns of the reviewers, and to improve the presentation by focusing on reducing the confusion introduced by the terminology.

**Justification For Why Not Lower Score:**

N/A

---

### Decision · Program_Chairs · 2024-01-16

Reject